# Two-Hop Monitoring Mechanism Based on Relaxed Flow Conservation Constraints against Selective Routing Attacks in Wireless Sensor Networks

**DOI:** 10.3390/s20216106

**Published:** 2020-10-27

**Authors:** Abdelouahid Derhab, Abdelghani Bouras, Mohamed Belaoued, Leandros Maglaras, Farrukh Aslam Khan

**Affiliations:** 1Center of Excellence in Information Assurance (CoEIA), King Saud University, Riyadh 11451, Saudi Arabia; fakhan@ksu.edu.sa; 2Department of Industrial Engineering, College of Engineering, Alfaisal University, Riyadh 11533, Saudi Arabia; abouras@alfaisal.com; 3LICUS Laboratory, Department of Computer Science, University of 20 August 1955, Skikda 21000, Algeria; m.belaoued@univ-skikda.dz; 4School of Computer Science and Informatics, De Montfort University, Leicester LE1 9BH, UK

**Keywords:** intrusion detection system, wireless sensor network, selective routing attack, two-hop monitoring, relaxed flow conservation constraint

## Abstract

In this paper, we investigate the problem of selective routing attack in wireless sensor networks by considering a novel threat, named the upstream-node effect, which limits the accuracy of the monitoring functions in deciding whether a monitored node is legitimate or malicious. To address this limitation, we propose a one-dimensional one-class classifier, named relaxed flow conservation constraint, as an intrusion detection scheme to counter the upstream node attack. Each node uses four types of relaxed flow conservation constraints to monitor all of its neighbors. Three constraints are applied by using one-hop knowledge, and the fourth one is calculated by monitoring two-hop information. The latter is obtained by proposing two-hop energy-efficient and secure reporting scheme. We theoretically analyze the security and performance of the proposed intrusion detection method. We also show the superiority of relaxed flow conservation constraint in defending against upstream node attack compared to other schemes. The simulation results show that the proposed intrusion detection system achieves good results in terms of detection effectiveness.

## 1. Introduction

A Wireless Sensor Network (WSN) [1] is a set of tiny sensor nodes, which are resource-constrained in terms of energy, bandwidth, processing, and storage capacities. The main task of a WSN is to collect/aggregate data from the sensor nodes to a base station (or sink) using a hop-by-hop communication. The data collection can be either event-driven or periodic-based. In the event-driven class, data are delivered to the sink after the occurrence of events. In the periodic-based class, each sensor node periodically sends its measurement towards the sink.

Sensor nodes could be compromised and controlled by an attacker. For example, an adversary can perform node capture attack by physically accessing the sensor node and uploading a malicious code to launch different types of attacks [2,3]. To avoid this situation, some preventive security mechanisms such as authentication, cryptography, and key management could be implemented [4,5]. However, it is necessary to deploy intrusion detection systems (IDSes) [6,7,8,9] to deal with other non-preventive attacks, such as selective routing/forwarding attack, where a malicious node refuses to forward some/all data packets that it receives. Some of the intrusion detection schemes that are proposed to deal with this attack use the watchdog principle, in which a node *u* that sent a packet to node *v* can overhear whether *v* has forwarded the packet to its downstream neighbor (or next hop) along the path towards the sink or not.

This attack has been tackled in many research studies [10,11,12,13,14,15,16,17,18,19,20,21,22,23]. The simplest way to detect a selective routing attack is to use the watchdog technique. This istechnique not sufficient for detecting the selective routing attack as the leaf nodes, for example, in the tree-based topology are not monitored. Hence, it is important to increase the coverage of monitoring by including the upstream neighbors (previous hop). In addition, in some routing schemes [24,25,26], a node that is at *l*-hop away from the sink can choose its next hop from *l*-hop, (l+1)-hop, and (l−1)-hop away from the sink node. Thus, it is important for each node to monitor all its neighbors to make a correct routing choice.

The role of an IDS is to identify the malicious nodes. The detection effectiveness of the IDS depends on the features that are used to describe the node’s normal behavior. If the feature with respect to the monitored node exceeds a defined threshold, then the node is considered as malicious. However, adopting some features might mislead the detection system, and make it falsely accuse a legitimate node as malicious. Figure 1 clearly explains this situation. In the figure, each node monitors the packet receiving rate of its upstream neighbor along the path toward the sink *B*. The value above each link (a,b) indicates the sending rate (resp., receiving rate) to *b* (resp., from *a*). Figure 1a shows the normal traffic rates of the network. Figure 1b shows the state of the network when nodes *D*, *H*, and *K* become compromised and start behaving maliciously by dropping some packets. As *D*, *H*, and *K* reduce their sending rates, their respective downstream neighbors *I* and *L* also have to reduce their sending rates accordingly. As a result, node *B* will falsely accuse nodes *I* and *L* of being malicious and, hence, a false positive is recorded. This attack is called upstream-node effect, i.e., node is falsely accused as malicious because its upstream node is behaving maliciously.

In this paper, we propose an intrusion detection system, which aims to detect the selective routing attack, prevents the upstream-node effect, and ensures that each node can monitor all its neighbors. The main contributions of the paper are the following:We propose a one-dimensional one-class classifier, named relaxed flow conservation constraint, as an intrusion detection scheme to counter the upstream node effect. It determines a boundary (i.e., threshold), which separates between normal packet loss and packet loss due to attacks. Instead of building the classifier using constrained optimization problem, as in one-class SVM [27], the proposed classifier has two main characteristics: (1) it incurs less computational cost and (2) it is most appropriate for networks that are operating under quasi-stable conditions in terms of link quality. In this case, the instances that compose the normal training data set are close to each other and can be grouped into one small bounded interval.We use four variants of the relaxed flow conservation constraint, depending on the type of monitored node. Three constraints are applied using one-hop knowledge, and the fourth one is applied while using two-hop information. To obtain the two-hop information, we propose the two-hop energy-efficient reporting scheme, which is complemented with some security mechanisms, such as authentication, encryption, filtering of ratings, and isolation.We analyze the security of the proposed IDS as well as its resilience probability against selective routing attacks and unfair ratings. Contrary to the state-of-the art methods, we show that the relaxed flow conservation constraint prevents the upstream node effect. We also analytically and experimentally evaluate the performance of the proposed IDS-based relaxed flow conservation constraints.

The rest of the paper is organized, as follows: Section 2 presents the related work. Section 3 provides the system model and problem statement. In Section 4, we justify our design choices. Section 5 describes the intrusion detection system that is based on the relaxed flow conservation constraints. In Section 6, we analyze the security of the proposed IDS against some attacks as well as its resilience probability against selective routing attacks and unfair ratings. In Section 7, we evaluate the resiliency of the relaxed flow conservation constraint against upstream-node effect and then compare it with two state-of-the-art features. The analysis of performance complexity, and node lifetime estimation are provided in Section 8. In Section 9, we present the simulation results. Finally, Section 10 concludes the paper.

## 2. Related Work

### 2.1. Intrusion Detection Schemes for Selective Routing Attacks

The selective routing attack has been widely studied in the literature. In order to counter the selective routing attack, Karlof et al. [28] suggested sending the same packet on a set of disjoint paths. However, this method shows poor security resilience if there is at least one compromised node along each disjoint path. It also incurs high communication overhead and high energy consumption when the number of paths increases.

In [11], each node counts the number of packets not forwarded by its downstream node. If this number exceeds a given threshold, then an alarm is raised. In [10], each node counts the number of packets not forwarded by its neighbor node within a time window; if the majority of monitoring nodes produce an alarm regarding the same node, the latter will be considered to be malicious. This method incurs a communication cost of O(N), where *N* is the number of nodes in the network. In [23], each node calculates the received power rate and the arrival packet rate of its upstream neighbor.

Stetsko et al. [12] proposed an intrusion detection system based on collaboration between neighbor nodes. They built a profile of the monitored node that consists of two features: (i) packet dropping rate, defined as: the number of packets sent to a certain node during a predefined period of time but not forwarded by that node and (ii) packet forwarding rate, defined as the number of packets received from its neighbors and consequently forwarded to its parent node during a predefined period of time. Each measured profile is then broadcast to its neighbors. In this way, the monitoring node can have different records on the same monitored node and can select the most accurate one. The main drawback is that it is not explained how accurate records are selected and how it is possible to detect unfair records provided by dishonest nodes.

Yu et al. [13,17] proposed a scheme that chooses some nodes as checkpoints along the path. When a checkpoint node receives a packet, it sends an acknowledgment (ACK) packet to the source node via its upstream neighbor. If any intermediate node does not receive enough ACK packets from its downstream neighbor during a time period, then it suspects that the packets have been dropped by a downstream malicious node. In addition, each node uses a one-way hash key chain to ensure the authenticity of packets. The problem with this scheme is that it consumes considerable energy on sending acknowledgement packets and it does not seem suitable for resource-constrained WSNs. If *L* is the average path length, then there is a need to forward O(|CHK|L) ACK packets for each data packet, where |CHK| is the number of checkpoint nodes along a path. If *L* is proportional to $\sqrt{N}$ and *m* data packets are generated during each time period, then the overall message complexity of this scheme for *N* nodes is $O(|CHK|\times m\times N^{\frac{3}{2}})$/time period. In addition, it might detect the existence of malicious activities but cannot identify exactly which node is malicious.

Brown and Du [16] proposed a cluster-based scheme in order to detect selective routing attack using a heterogeneous WSN consisting of a few high-end sensor nodes (H-sensors or cluster heads) and a large number of Low-end sensor nodes (L-sensors). The detection procedure is as follows: each monitoring node passively listens to its downstream node. If the latter drops a packet, then the monitoring node (L-sensor) will include the ID of the dropper node in a packet and send it to the cluster head via another route. Based on the reports, the cluster head performs the sequential probability ratio test, which calculates the percentage of dropped packets in all forwarding packets, to check whether the L-sensor is compromised or not. This scheme has the following drawbacks: (1) it suffers from the single node failure problem when the cluster head is compromised and (2) no reliable mechanism is proposed to retransmit the dropped packets.

Li et al. [18] proposed a detection scheme based on the sequential mesh test method [29]. In [18], the network is organized into clusters, and any node that does not observe the forwarding of its downstream neighbor sends a report packet to the cluster head. The latter runs a sequential mesh test, which extracts a small quantity of samples to run the test, instead of regulating the total time of test in advance. This scheme also suffers from the single node failure problem.

Xin-sheng et al. [15] proposed a detection scheme based on a hexagonal WSN mesh topology. This scheme uses the nodes, which are capable of monitoring the forwarding of two successive nodes in the routing path. When these monitoring nodes detect selective routing attack, they resend the dropped packets to the destination node. The problem with this scheme is that it depends on a static and specific kind of topology. In addition, there is no countermeasure mechanism when the monitoring node is compromised.

Kaplantzis et al. [14] proposed a centralized intrusion detection scheme that is based on One-class Support Vector Machines (OSVMs). The detection process is performed by the sink and uses two features, the incoming bandwidth, and the hop count taken by each packet to reach the sink node, in order to detect selective routing and black hole attacks. If the observed behavior is different from the trained one, then a detection alarm is raised.

Jamshidi et al. [30] proposed a distributed lightweight method based on learning automata in order to select the most secure routes for forwarding packets, and thus avoiding those that contain malicious nodes. Fu et al. [31] proposed data clustering algorithm to detect and isolate malicious cluster heads that launch selective routing attacks. Mehetre et al. [32] introduced a trustable secure routing method for clustered WSNs that can also offer security through encryption and verification of packets on each source node. The proposed method offers advanced security, privacy, and energy efficiency features, but it is a rather heavyweight method for WSNs both in terms of computational overhead and increased complexity.

### 2.2. Discussion and Desired Design Principles

We can observe that the majority of schemes that are against selective routing attacks perform a downstream monitoring, which does not allow for a node to ensure monitoring coverage of all its neighbors. We can also observe that the majority of works use the following features to describe the node’s behavior: packet dropping rate, packet forwarding rate, packet arrival rate, and number of Ack packets (i.e., number of received packets). In Section 7, we will show that these features do not prevent the upstream-node effect and, hence, they incur poor detection accuracy and a high false positive rate. Thus, an IDS against the selective routing attack should have the following properties:Effectiveness: the IDS should provide high detection rate while providing low false positive rate. This aim is ensured by using path-free features to describe the node’s behavior.Full network monitoring coverage: the IDS should be able to monitor the whole network. To achieve this, each node should include downstream and upstream monitoring processes.Localized and fully-distributed: the IDS should be localized, i.e., the tasks of each node, e.g., data acquisition and data analysis should be based solely on the knowledge of its local neighborhood. The watchdog technique, in which a node listens promiscuously to packets passing by its neighborhood, is a good example of a localized solution as it does not incur any additional communication overhead. Additionally, the IDS should be distributed across different nodes, and does not rely on a centralized entity to avoid the single-point-of-failure problem.Minimized resource consumption: the IDS should be designed in a way that it can be lightweight and implementable in the resource-constrained sensor nodes by using only a small number of resources. Additionally, it is important that the IDS incurs less communication overhead to reduce energy consumption and prolong the node and network lifetime.

## 3. System Model and Problem Statement

Given a set of *N* sensor nodes and a sink node, the network is structured as a tree-topology rooted at the sink. At every time interval Δt, each node *i* sends a data packet along the tree towards the sink node, and monitors a set of nodes $W_{i}\subseteq L_{i}$, such that $L_{i}$ denotes the set of *i*’s neighbors. We consider two forms of the selective routing attack:An attacker *i* drops some/all data packets coming from all its upstream neighbors.An attacker stops generating its own data packets.

At every time interval $\Delta_{j}=\Delta$, called the monitoring period, each node *i*, which monitors node *k*’s behavior, constructs a vector (or profile) $a_{j}^{ik}\in N^{d}$ that is composed of *d* attributes $\left\{a_{jl}^{ik}:l=1\ldots d\right\}$. The attributes can denote: the number of dropped packets, the number of received packets, the number of generated packets, etc. To be able to perform monitoring, $\Delta_{j}$ must be larger than Δt. We remove the superscripts *i* and *k* to ease the notation burden.

After a time period *T*, each sensor node has collected a set of *n* d-attribute profile vectors $a_{j}=\left\{a_{jl}:l=1\ldots d\right\}$, called the training data set. The latter can also be written as d×n matrix $\left[a_{jl}\right]$, where $a_{jl}$ is the number of occurrence of feature *l* in profile $a_{j}$ and *n* is the number of profiles. A profile $a_{j}$ can be a one-attribute vector that records, for example, the number of packets sent/forwarded by the monitored node during the monitoring period $\Delta_{j}$.

During the testing phase, the intrusion detection problem is to check at each $\Delta_{j}$ (such that j>n) if the observed profile $a_{j}$ is normal or not.

## 4. Design Choices

### 4.1. One-Class Classifier

Statistical-based and rule-based approaches [10,33,34] use some thresholds to distinguish between normal and abnormal behaviors. These thresholds depend on some physical parameters, like signal noise and other radio parameters that vary from a network deployment scenario to another. Hence, it is not feasible to calculate the required threshold for each network deployment and upload the corresponding binary code to the sensor node.

The one-class classifier (OCC) assumes that all of the training instances belong to one class, named positive class. The negative class, which represents abnormal behavior is absent or unknown. The aim of the classifier is to find a separating boundary between the normal instances and the rest of instances [35]. The OCC is solved using either density estimation or boundary description methods. In the density estimation method, an object is considered to be non-positive (or outlier) when the object falls into a region with a density lower than some threshold value. However, this method requires large data sets [36]. In the boundary description method, the classifier sets a boundary that encompasses almost all of the positive points with the minimum radius. Any test instance that does not fall within the learned boundary is declared as anomalous. The main issue that arises is when the training data set is composed of different regions with different density levels. Instances in low-density regions will be rejected although they are normal. Additionally, when the boundary of the dataset is long and non-convex, the required number of training instances is likely to be very high [35]. Thus, the one-class classifier gives its best results in terms of detection rate and false positive rate when the training instances are close to each other and can be grouped into one small region. The question that might arise is whether this small region can be constructed in WSNs or not. To answer this question, we are going to state the following observations and results.

Lin et al. [37] experimentally studied the stability of a testbed WSN. In this study, the transmissions are scheduled to avoid collisions and it is observed that the packet reception ratio of stable links (i.e., link quality remains constant at a certain level) are experiencing small fluctuations that are mainly caused by multi-path fading of wireless signals. According to this study, this case occurs, especially at night when there is no human movement or Wi-Fi interference.

DOZER [38], KOALA [39], and DISSENSE [40] are three periodic-based data collection protocols that are designed for long-term monitoring applications in WSNs, and have been implemented and tested on real platforms. DOZER and DISSENSE provide a packet delivery ratio of 98–99%. KOALA, on the other hand, achieves a ratio of 99.99%. This implies that the average packet reception ratio of a link is also lower-bounded by 98–99%.

The above results show that the instances comprising the normal traffic behavior (i.e., number of received packets) are close to each other and, hence, it is possible to build a small bounded region composing all these instances. This implies that the choice of one-class classifier here is appropriate and justified to monitor stable links or WSNs under quasi-stable conditions (i.e., constant flow rate, collision-free transmissions, no mobility, no signal interference, …, etc). The boundary, which separates between normal traffic and attack, is self-learned from each deployment setting, and it does not depend on a predefined threshold as in the statistical-based and rule-based approaches. In this situation, we adopt the one-class classifier to train the intrusion detection system.

### 4.2. Path-Free Features

The feature is a quantitative metric used by the monitoring node to evaluate the monitored node’s behavior. As our objective is to design an IDS that prevents the upstream-node effect, the computation of the node’s feature must not depend on the features that are related to its upstream node. Thus, we classify the features used to monitor the node’s behavior as path-dependent and path-free.

A feature is called path-dependent if the feature value assigned to a node depends on the feature value assigned to its upstream neighbor. On the other hand, a feature is called path-free if the feature value that is assigned to a node does not depend on the feature value assigned to its upstream neighbor.

Formally, we consider that a data flow is routed along the path [a,⋯,j,k,l,⋯,b]. Each node *l* along the path monitors the activity of its upstream node *k* and measures the corresponding feature $ft_{kl}$. A feature is called path-dependent if any increase (resp., decrease) in $ft_{jk}$ will increase (resp., decrease) $ft_{kl}$. A feature is called path-free if the variation of $ft_{kl}$ does not depend on $ft_{jk}$. For example, the packet sending rate is a path-dependent feature, as its variation at a given node affects its downstream node. Although the packet dropping and forwarding rates are considered to be path-free according to this definition, they cannot prevent the upstream-node effect, as shown in Section 7.

## 5. Intrusion Detection Scheme Based on Relaxed Flow Conservation Constraints

As stated in Section 2, the intrusion detection systems for selective routing attacks should at least use (1) downstream and upstream monitoring processes, and (2) path-free features to describe the network traffic. For this purpose, we propose path-free features that are based on the relaxed flow conservation constraint [41].

### 5.1. One-Dimensional One-Class Classifier: Relaxed Flow Conservation Constraint

We model WSN as a directed tree G=(V,E), where each edge e=(u,v) has a defined capacity denoted by C(e) or C(u,v). Each node *m* monitors its downstream neighbor (i.e., *m*’s parent denoted by $Parent_{m}$) and all of its upstream neighbors (i.e., *m*’s children denoted by $Child_{m}$). Based on the known multi-flow problem [42], we define a flow as a function f(u,v):V×V→N, which counts the number of packets sent from node *u* to node *v* during a monitoring period of length Δ and satisfies the following constraints:Flow conservation constraint: the sum of flows entering *v* and that generated by *v* during a monitoring period $\frac{\Delta }{\Delta t}$ must equal the flow leaving *v* towards $Parent_{v}$ for all nodes, except the sink node and the leaf nodes in the tree-based topology, as depicted in Figure 2. Formally:
$$\left(\sum_{\omega \in Child_{v}}f\left(\omega ,v\right)+\frac{\Delta }{\Delta t}\right)-f\left(v,Parent_{v}\right)=0$$A flow f(u,v) can also be written as:
$$f\left(u,v\right)={self}_{u}+\sum_{\omega \in Child_{u}}g\left(\omega ,u,v\right)$$
such that: ${self}_{u}$ is the flow actually generated by node *u*, and g(ω,u,v) is the flow received by node *u* from node ω and forwarded later to node *v*.Capacity constraint: the flow along any edge must be positive and less than the capacity of that edge. Formally: 0≤f(u,v)≤C(u,v).

As the wireless channel is error-prone, some packets could be lost due to collisions, and thus the flow conservation constraint cannot be ensured. To deal with this issue, we use instead a relaxed flow conservation constraint, which represents the difference between (a) the flows leaving *v*, and (b) the sum of flows entering *v* and the flow generated by *v*. The value of this difference is bounded between 0 and a given threshold $\in N^{*}$. Our objective is to determine the threshold that encompasses all the normal instances $d_{v}$ such that: $$d_{v}=\left(\sum_{\omega \in Child_{v}}f\left(\omega ,v\right)+\frac{\Delta }{\Delta t}\right)-f\left(v,Parent_{v}\right)$$

To this end, the relaxed flow conservation constraint can be written as one-dimensional one-class classifier that satisfies the following:$$d_{v}\leq \max_{t\in [0,T]}\delta_{v}\left(t\right)=\delta_{v}$$
where $\delta_{v}\left(t\right)$ denotes the flow drop, defined as the number of packets sent and forwarded by node *v* and lost due to normal failures during a monitoring period *t*. $\delta_{v}$ is the highest value of $d_{v}$ observed during the training phase *T*. Formally, we have:$$d_{v}=\left(\sum_{\omega \in Child_{v}}f\left(\omega ,v\right)+\frac{\Delta }{\Delta t}\right)-f\left(v,Parent_{v}\right)\leq \delta_{v}$$

The above constraint can also be written as: $$d_{v}=\left(\sum_{\omega \in Child_{v}}f\left(\omega ,v\right)+\frac{\Delta }{\Delta t}\right)-\left({self}_{u}+\sum_{\omega \in Child_{v}}g\left(\omega ,v,Parent_{v}\right)\right)\leq \delta_{v}$$

**Proposition** **1.**
*$d_{v}$ is a path-free feature.*


**Proof.** Let us consider the path $u⟶v⟶Parent_{v}$. Let *u* be a malicious node, which drops some packets. A feature is called dependent-path if and only if any increase in $d_{u}$ will eventually increase $d_{v}$.Let $f^{t}\left(u,v\right)$, $d_{v}^{t}$, $self_{v}^{t}$, and $g^{t}\left(u,v,Parent_{v}\right)$ be the value of f(u,v), $d_{v}$, $self_{v}$ and $g(u,v,Parent_{v})$ at time *t*. $d_{v}^{t}$ can be written as: $\left(\sum_{\omega \in Child_{v}}f^{t}\left(\omega ,v\right)+\frac{\Delta }{\Delta t}\right)-f^{t}\left(v,Parent_{v}\right)$. At time $t^{\prime }>\left(t+\Delta \right)$ node *u* starts behaving maliciously by dropping some/all packets, its $d_{u}^{t^{\prime }}>\delta_{u}$, and it can be written as: $\left(\sum_{\omega \in Child_{u}}f^{t^{\prime }}\left(\omega ,u\right)+\frac{\Delta }{\Delta t}\right)-f^{t^{\prime }}\left(u,v\right)$. $f^{t^{\prime }}\left(u,v\right)$ is composed of two terms: $self_{u}^{t^{\prime }}$ and $\sum_{w\in Child_{u}}g^{t^{\prime }}\left(w,u,v\right)$. When *u* is acting maliciously, $self_{u}^{t^{\prime }}<self_{u}^{t}$ and/or $\sum_{\omega \in Child_{u}}g^{t^{\prime }}\left(\omega ,u,v\right)<\sum_{\omega \in Child_{u}}g^{t}\left(\omega ,u,v\right)$, and hence $d_{u}^{t^{\prime }}>d_{u}^{t}$.Node *v* is behaving normally at time *t* and $t^{\prime }$. Thus, $d_{v}^{t}=\left(\sum_{\omega \in Child_{v}}f^{t}\left(\omega ,v\right)+\frac{\Delta }{\Delta t}\right)-f^{t}\left(v,Parent_{v}\right)<\delta_{v},\mathrm{and}\;$
$d_{v}^{t^{\prime }}=\left(\sum_{\omega \in Child_{v}}f^{t^{\prime }}\left(\omega ,v\right)+\frac{\Delta }{\Delta t}\right)-f^{t^{\prime }}\left(v,Parent_{v}\right)<\delta_{v}$.As $f^{t^{\prime }}\left(u,v\right)<f^{t}\left(u,v\right)$, $\sum_{w\in Child_{v}}f^{t^{\prime }}\left(w,v\right)<\sum_{\omega \in Child_{v}}f^{t}\left(\omega ,v\right)$. In addition, node *v* will send at time $t^{\prime }$ less packets compared to time *t*.Formally, $f^{t^{\prime }}\left(v,Parent_{v}\right)<f^{t}\left(v,Parent_{v}\right)$ as:
$\begin{array}{c} self_{u}^{t^{\prime }}=self_{u}^{t},\mathrm{and}\;\\ \sum_{\omega \in Child_{v}}g^{t^{\prime }}\left(\omega ,v,Parent_{v}\right)<\sum_{w\in Child_{v}}g^{t}\left(w,v,Parent_{v}\right) \end{array}$
At time $t^{\prime }$, there will be less traffic flow in the neighborhood of node *v* when compared to time *t*, and we will obviously observe less channel contentions and less packet collisions, which decreases the flow drop. Formally, $d_{v}^{t^{\prime }}<d_{v}^{t}$. As a result, an increase in $d_{u}$ has not made *v* increase its $d_{v}$ and, hence, $d_{v}$ is a path-free feature. □

Based on the above constraint, we derive the three following monitoring detection constraints (or path-free features) C1, C2, and C3 used by the monitoring node *m* to monitor its downstream node *v* and its upstream node *u* (as shown in Figure 2).
$$\begin{array}{cc} C1: & f\left(m,v\right)-g\left(m,v,Parent_{v}\right)=d_{v}^{1}\leq \varphi_{m}^{v}\\ C2: & \frac{\Delta }{\Delta t}-self_{u}=d_{u}^{2}\leq \phi_{m}^{u}\\ C3: & \sum_{\omega \in Child_{u}}f\left(\omega ,u\right)-\sum_{\omega \in Child_{u}}g\left(\omega ,u,m\right)=d_{u}^{3}\leq \psi_{m}^{u} \end{array}$$

$\varphi_{i}^{j}$, $\phi_{i}^{j}$, and $\psi_{i}^{j}$ denote the thresholds used by the monitoring node *m* to check constraint (C1), constraint (C2), and constraint (C3) satisfaction with respect to the monitored downstream node *v* and upstream node *u*. $\varphi_{m}^{v}$, $\phi_{m}^{u}$, and $\psi_{m}^{u}$ are the highest values of $d_{v}^{1}$, $d_{u}^{2}$, and $d_{u}^{3}$, respectively, observed by the monitoring node *m* during the training phase.

C1 (or one-hop downstream monitoring and detection process) can be applied by each sensor node except for the sink node. Node *m*, using the watchdog technique, checks if the data packets that it has sent to its parent node *v* are forwarded by the latter. If the difference between the two elements, f(m,v) and $g(m,v,Parent_{v})$, exceeds the threshold $\varphi_{m}^{v}$, node *v* will be considered malicious. Otherwise, it is normal.

C2 (or one-hop upstream monitoring and detection process) can be applied by each sensor node, except for the leaf nodes in the tree-based network topology. As each node *u* (except the sink node) has to generate $\frac{\Delta }{\Delta t}$ packets during each monitoring period and send them to its parent node. Node *m* checks if the number of packets $self_{u}$ received from its upstream node *u* is less than $\left(\frac{\Delta }{\Delta t}-\phi_{m}^{u}\right)$. If so, node *u* is considered to be malicious. If node *u* is a leaf node, this judgment is sufficient for node *m*. Otherwise, node *m* still cannot judge node *u* and it needs to check constraint (C3).

C3 (or two-hop upstream monitoring and detection process) can be applied by each sensor node except for the leaf nodes as well as their parents in the tree-based network topology. The monitoring node *m* checks whether the data packets, where node *u* received from its children, are forwarded by node *u* to node *m*. If the difference between the two elements $\sum_{\omega \in Child_{u}}f\left(\omega ,u\right)$ and $\sum_{\omega \in Child_{u}}g\left(\omega ,u,m\right)$ exceeds the threshold $\psi_{m}^{u}$, node *u* will be considered as malicious. Otherwise, it is normal. We can notice that all the terms of constraint (C3) are one-hop information and can be directly obtained by node *m*, except for $\sum_{\omega \in Child_{u}}f\left(\omega ,u\right)$. The issue here is how to report this two-hop information to node *m* so that constraint C3 can be verified.

### 5.2. Two-Hop Energy-Efficient and Secure Reporting Scheme

In order to report information to a two-hop neighbor (i.e., grandparent in the tree), the trivial solution is that each node $\omega \in Child_{u}$ acts as a witness and sends a packet containing f(ω,u) to the monitoring node *m* through an intermediate node. However, a malicious intermediate node can simply drop this packet and prevent node *m* from receiving it. The other solution is to use an adaptive power transmission that can reach the two-hop neighbors by using higher power transmission level. However, this solution exhausts the node’s battery faster than the normal power and makes the packet contend with more transmitting nodes.

To address the above drawbacks, we propose an energy-efficient and secure method, allowing for nodes that can monitor the flows entering and leaving *u* to report their observations to the monitoring node *m*.

#### 5.2.1. Witness Set Model

When the monitoring node *m* cannot judge directly based on its own observations whether the behavior of its upstream node *u* is normal or not, it has to collect ratings on node *u* from other nodes, called the witness nodes. Witness nodes that can rate *u* are: (1) nodes, which can overhear the transmission of node *u* and at least one of *u*’s children and (2) *u*’s children.

To define the set of witness nodes, as denoted by $WN_{u}$, which can rate node *u*, we have to define first the witness region for the transmitter relay pair (ω,u) as: $$WR_{\omega \to u}=\left\{p=\left(x,y\right)\in R^{2}:dis\left(p,\omega \right)\leq r\wedge dis\left(p,u\right)\leq r\right\}$$
where *r* denotes the node transmission radius, and $dis(p_{1},p_{2})$ denotes the euclidean distance between two points $p_{1}$ and $p_{2}$. Figure 3a shows the witness region for $\left(u_{1},u\right)$ as shaded region. The total witness region for node *u*, which is shown in Figure 3b, is defined by $WR_{u}=\underset{\begin{array}{c} \omega \in Child_{u} \end{array}}{⋃}WR_{\omega \to u}$.

The set of witness nodes $WN_{u}$ are the nodes that are located in the witness region $WR_{u}$, except for node *u*. Let us consider that a witness node *w* can overhear the transmission between a set of nodes $Child_{u}^{\left[w\right]}\subseteq Child_{u}$ and node *u*. The witness node *w* applies the following constraint C4 to rate node *u*.
$$C4:\;wit_{w}^{u}=\sum_{\omega \in Child_{u}^{\left[w\right]}}f\left(\omega ,u\right)-\sum_{\omega \in Child_{u}^{\left[w\right]}}g\left(\omega ,u,Parent_{u}\right)\leq \chi_{w}^{u}$$

C4 (or one-hop witness monitoring and detection process) node *w* checks if the data packets, which were sent by nodes $\in Child_{u}^{\left[w\right]}$ to node *u*, are forwarded by node *u*. If the difference between the two elements: $\sum_{\omega \in Child_{u}^{\left[w\right]}}f\left(\omega ,u\right)$ and $\sum_{\omega \in Child_{u}^{\left[w\right]}}g\left(\omega ,u,Parent_{u}\right)$ exceeds the threshold $\chi_{w}^{u}$, node *u* will be considered as malicious. Otherwise, it is normal. $\chi_{w}^{u}$ is the highest value of $wit_{w}^{u}$ recorded by node *w* while observing node *u* during the training phase.

If a witness node gives fair rating, then it is called honest. Otherwise, it is called liar. We can notice that each witness node located in $WR_{u}$ partially monitors the traffic flow coming to node *u*. As $Child_{u}^{\left[w\right]}\neq Child_{u}^{\left[w^{\prime }\right]}$, the amount of traffic monitored by *w* differs from that of *w*’ and, hence, it is impossible to fairly compare between two ratings $wit_{w}^{u}$ and $wit_{w^{\prime }}^{u}$ that resulted from different observations and tell which one is honest or liar. Figure 3 shows that the nodes, which are witnessing node *u*, are {w1,w2,u1,u2,u3}. Nodes w1, u1, and u2 can only monitor the traffic flows (u1,u) and (u2,u). On the other hand, nodes w2, u1, and u3 can only monitor the traffic flows (u1,u) and (u3,u). To deal with this issue, we use binary values to rate node *u*. Formally, the reputation rating held by witness node *w* about node *u*, denoted as $rating_{w}^{u}$, is set to 1 if *u* is normal and 0 otherwise.

#### 5.2.2. Two-Hop Energy-Efficient Reporting Scheme

We propose a lightweight method for reporting node *w*’s opinions about node *u* to node $Parent_{u}=m$. Each node *i* stores the following variables: (1) $Child_{i}$ (i.e., the set of *i*’s upstream nodes in the tree-based routing topology), (2) $Witnessing_{i}$ (i.e., the set of witness nodes that can monitor any node in $Child_{i}$), (3) $Witnessed_{i}$ (i.e., the set of nodes that node *i* can monitor, ordered according to the lexicographical order (where 0<1<2…)), (4) $Relay_{i}$ (i.e., the set of witness nodes that use *i* as a relay to reach their two-hop monitoring nodes). The method is executed in two phases: (a) initialization phase, executed once, and (b) a rating phase, executed periodically. Figure 4 shows the value of some variables that resulted from executing the reporting scheme. In the figure, the directed links represent the tree-based routing topology, and a dotted line drawn from a node *w* to link (ω,u) indicates that *w* can overhear the messages transmitted by ω and *u*.

In the initialization phase, the following operations are carried out:Each witness node *w* that can observe the network traffic traversing the link (ω,u) adds *u* to the set $Witnessed_{w}$. Subsequently, *w* broadcasts $Witnessed_{w}$ with TTL=2.When a monitoring node *m* receives $Witnessed_{w}$ and $W_{w}=\left\{Child_{m}\cap Witnessed_{w}\right\}\neq Ø$ (i.e., *w* can monitor at least one of *i*’s children), *m* adds *w* to $Witnessing_{m}$. Subsequently, it stores for each $p\in W_{w}$ the position of *p* in $Witnessed_{w}$. Afterwards, node *m* broadcasts a message containing $Witnessing_{m}$. In Figure 4, as node 3 can observe node *m*’s children, it is added to $Witnessing_{m}$.When a node *r* receives $Witnessing_{m}$, it checks for all $j\in Witnessing_{m}$ if $\left(j\in N_{r}\wedge j\notin N_{m}\right)$ (i.e., witness node *j* cannot reach *m* directly, but only through *r*). If so, it adds *j* to $Relay_{r}$.

In the rating phase, each node *i* has to report periodically its observations and forward the observations of nodes in $Relay_{i}$ to the monitoring nodes (i.e., $\left(|Relay_{i}|+1\right)$ observations are periodically reported by node *i*). Each observation from node *w* has to include the ID of all nodes $j\in Witnessed_{w}$ and their ratings. As *N* is the total number of nodes, the ID and the rating can be encoded using $\log_{2}N$ bits and one bit, respectively. Thus, the size of this observation is: $|Witnessed_{w}|\left(\log_{2}N+1\right)$ bits, and the total size of information that is transmitted periodically by each node *i* is $(\sum_{w\in Relay_{i}}|Witnessed_{w}|+|Witnessed_{i}\left|\right)\left(\log_{2}N+1\right)$ bits. $|Relay_{i}|$ and $|Witnessed_{w}|$ are upper-bounded by $d_{max}$, where $d_{max}$ is the maximum node degree of the network. As, for example, let us consider N=1024 and $d_{max}=10$, the size of one observation is 110 bits, and the total size of observations to be sent by node *i* is: 1210 bits. To reduce this size, we apply a spatial-temporal compression, as follows:Spatial compression: as the monitoring node knows the position of its children in $Witnessed_{w}$, we do not need to transmit the couple $\left(j,rating_{w}^{j}\right)$ for each $j\in Witnessed_{w}$. Instead, *w*’s observation will consist of *w*’s ID, and a boolean bit vector, denoted by $Rep_{w}$ and composed of $|Witnessed_{w}|$ bits. $Rep_{w}\left[i\right]=1$ if $rating_{w}^{j}=1$ (i.e., node *j*, which is at position *i* in $Witnessed_{w}$, is rated normal by *w*, and 0 otherwise). In Figure 4, node *m* knows that the third and the fourth elements in $Rep_{3}$ correspond to the ratings of its children 11 and 12, respectively. The size of one observation is reduced to $\left(\log_{2}N+|Witnessed_{w}|\right)$ bits. The above example gives 20 bits per observation.Temporal compression: instead of transmitting all of the observations in one message, each node *i* piggybacks each observation to one beacon message. To do so, it defines a time window $TR_{i}$ composed of $TB\times (|Relay_{i}|+2)$ time slots, where TB is the time between two beacon broadcasts. $|Relay_{i}|$ slots are used to relay the ratings sent by nodes in $Relay_{i}$. The last two time slots are used to send *i*’s observation: one encrypted and sent to other relay nodes to reach *i*’s two-hop monitoring nodes, and the non encrypted is broadcast by *i* to reach *i*’s one-hop monitoring nodes. The encryption mechanism is described in Section 5.2.3.

#### 5.2.3. Security Mechanisms for the Reporting Scheme

The observations that are exchanged among nodes are subject to many attacks, which aim at spoofing the source of observations and/or altering their contents. Thus, we enhance the two-hop energy-efficient reporting scheme by adding some security mechanisms.

In order to prevent the relay node *r* from altering the observations sent by *w*, each node *i* creates its two-hop broadcast key $THK_{i}$, which is a shared key between the node and its two-hop neighbors. This key is unknown for the one-hop neighbors. The creation of this key, as described in [43], is as follows: Each node *i* is preloaded with a transitory initial key $K_{IN}$ and a random number $RN_{i}$. It first computes its master key $MK_{i}=G\left(K_{IN},ID_{i}\right)$, where *G* is pseudo random function and $ID_{i}$ is node *i*’s identifier. Subsequently, it computes its two-hop broadcast key $THK_{i}=G(MK_{i},RN_{i}||RN_{i})$. Node *i* sends $RN_{i}$ encrypted with $K_{IN}$ to its two-hop neighbors. The latter use the received $RN_{i}$ to derive $MK_{i}$ and $THK_{i}$. We assume that an adversary can compromise a node only after $T_{min}$ time units from network deployment. Before the expiration of $T_{min}$, each node *i* erases $K_{IN}$ and $RN_{i}$ from its memory in order to prevent an adversary from compromising the keys of other nodes. Node *i* uses $THK_{i}$ to encrypt its observations.

An attacker might forge its identity to masquerade as another node and send fake observations. To allow for two-hop authentication of the observation, each node *i* generates a two-hop one-way hash chain (TOHC) $\left(H_{i}^{0},H_{i}^{1},H_{i}^{2},\cdots ,H_{i}^{K}\right)$. Each node within *i*’s two-hop neighborhood is preloaded with the last value ($H_{i}^{K}$). Upon generation of the *d*’th observation, node *i* appends ($H_{i}^{K-d}$) to $Rep_{i}$. Each monitoring node that receives the *d*’th observation can authenticate the source *i* by applying the one-way hash function on the received value $H_{i}^{K-d}$ and verifying whether the result is equal to the pre-loaded TOHC value $H_{i}^{K-(d-1)}$; $F\left(H_{i}^{K-d}\right)=H_{i}^{K-(d-1)}$.

When a node is physically captured, the adversary can retrieve all of the embedded security credentials and turn the node into a liar that fabricates fake reports, without being detected by the above security mechanisms. In the following subsections, we show how this issue is tackled under two models of selective routing attacks.

#### 5.2.4. First Selective Routing Attack Model

In this model, a malicious node is defined as the one behaving maliciously against all od its upstream neighbors. Let $\chi_{w}^{\omega \to u}$ denotes the highest flow drop observed on the link (ω,u) by the witness node *w* during the training phase. If node *u* is behaving maliciously against upstream neighbor ω, then constraint (C4) applied on link (ω,u) will be as follows:$$f\left(\omega ,u\right)-g\left(\omega ,u,Parent_{u}\right)>\chi_{w}^{\omega \to u}$$

**Proposition** **2.**
*Under the first selective routing attack model, any honest node located in $WR_{u}$ that rates u positively (resp., negatively), the other honest nodes located in $WR_{u}$ must also rate u positively (resp., negatively).*


**Proof.** If two witness nodes *w* and $w^{\prime }$ are monitoring node *u*, four cases can occurs:
$Child_{u}^{\left[w\right]}=Child_{u}^{\left[w^{\prime }\right]}$$Child_{u}^{\left[w\right]}\subset Child_{u}^{\left[w^{\prime }\right]}$$\left(Child_{u}^{\left[w\right]}\neq Child_{u}^{\left[w^{\prime }\right]}\right)\wedge \left(Child_{u}^{\left[w\right]}\cap Child_{u}^{\left[w^{\prime }\right]}=Ø\right)$$\left(Child_{u}^{\left[w\right]}\neq Child_{u}^{\left[w^{\prime }\right]}\right)\wedge \left(Child_{u}^{\left[w\right]}\cap Child_{u}^{\left[w^{\prime }\right]}\neq Ø\right)$
For all the four cases mentioned above, nodes *w* and $w^{\prime }$ apply constraint (C4) on node *u* as follows:
$$\sum_{\omega \in Child_{u}^{\left[w\right]}}f\left(\omega ,u\right)-\sum_{\omega \in Child_{u}^{\left[w\right]}}g\left(\omega ,u,Parent_{u}\right)>\sum_{\omega \in Child_{u}^{\left[w\right]}}\chi_{w}^{\omega \to u}=\chi_{w}^{u}$$
$$\sum_{\omega \in Child_{u}^{\left[w^{\prime }\right]}}f\left(\omega ,u\right)-\sum_{\omega \in Child_{u}^{\left[w^{\prime }\right]}}g\left(\omega ,u,Parent_{u}\right)>\sum_{\omega \in Child_{u}^{\left[w^{\prime }\right]}}\chi_{w^{\prime }}^{\omega \to u}=\chi_{w^{\prime }}^{u}$$We can notice that both nodes *w* and $w^{\prime }$ will reach the same opinion about node u’s reputation (i.e., normal or malicious) and, hence, two honest witness nodes rate the same node alike. □

From the above proposition, we can easily prove the following corollary:

**Corollary** **1.**
*Under the first selective routing attack model, if two nodes located in $WR_{u}$ are rating u differently, then one of them is liar and the other one is honest.*


A captured node can give positive and negative unfair ratings, which might jeopardize the whole reputation system and, thus, efficient protection against unfair rating is a basic requirement. The methods used to filter the ratings can be classified into two groups:Endogenous methods: they exclude unfair ratings by assuming that the ratings provided by honest witnesses are equal or statistically close to each other.Exogenous methods: they exclude unfair ratings by introducing external parameters like the reputation of the witness nodes. They are based on the assumption that the witnesses with low reputation are likely to give unfair ratings.

The exogenous approach is costly in terms of message overhead, as there is a need to collect witness nodes’ reputations from many sources. In the endogenous approach, the monitoring node *m* needs to get opinions from different *M* witness nodes about node *v*, which incurs a communication cost of O(M). On the other hand, exogenous approach require collecting reputation information on each witness node *w* to decide if *w*’s opinion will be excluded or not. If we consider that for each witness node, *K* nodes on average rate *w*. This leads to an overall complexity of O(K×M). Accordingly, we choose to use the endogenous approach. Node *m* checks constraint (C3) periodically by performing the following two phases. In the first phase, the monitoring node *m* compares its own rating about node *v* with the rating provided by a witness node *w* about the same node *v*. If the two ratings are different, *w* will be excluded from its list of two-hop honest witness nodes $Honest_{m}$, initialized to be the set of *m*’s two-hop neighbors. In the second phase, a monitoring node *m*, by applying constraint (C1) knows exactly how its downstream neighbor $Parent_{m}$ is behaving. If another node *w* gives a different opinion about $Parent_{m}$, node *m* can surely tell that *w* is a lying and it is excluded from $Honest_{m}$. At the end of each time period, node *m* calculates the sum of positive and negative ratings about all its upstream nodes *u*, denoted by $UpstreamPos_{i}^{u}$ and $UpstreamNeg_{i}^{u}$, respectively. As proved above, any two honest witness nodes that are monitoring the same transmitter relay pair (ω,u) report the same reputation rating. We assume that, for any upstream node *u*, the majority of nodes in $Honest_{m}\cap WN_{u}$ are honest. Except for leaf nodes, $\forall u:WN_{u}\neq Ø$ as $W_{u}$ contains *u*’s upstream nodes. The necessary conditions for node *m* to correctly rate its upstream node *u* are the following:There is at least one path with no liar or malicious nodes from each node in $\left(Honest_{m}\cap WN_{u}\right)$ to the monitoring node *m*.At least $\lfloor \frac{Honest_{m}\cap WN_{u}}{2}\rfloor +1$ nodes in $\left(Honest_{m}\cap WN_{u}\right)$ are honest.

#### 5.2.5. Second Selective Routing Attack Model

Under this model, a malicious node behaves maliciously against some of its upstream neighbors. Consequently, two honest witness nodes might not make the same judgment. For example, node 13 in Figure 4 can evade detection by behaving maliciously against node 8 but not against node 9. In this case, the witness nodes 7 and 10 will make different observations and, hence, node *m* cannot judge 13 correctly. If we assume that there is at least one honest node in $Witnessing_{m}$, a monitoring node *m* can judge its child *u* only when all nodes in $Witnessing_{m}$ report the same rating. Otherwise, node *m* does not take any decision. Instead, *u*’s upstream node, which considers *u* to be malicious by applying constraint (C1), stops routing data through *u* and switches to another parent node. If node *u* chooses to continue behaving maliciously against other upstream nodes, it will be isolated by these nodes and have no data to drop.

## 6. Security Analysis

### 6.1. Resilience against Some Attacks

In this section, we analyze the security of the two-hop reporting scheme, and discuss its resilience against some potential attacks.

Attacks against the key management system: if an adversary overhears exchanged messages during the key establishment phase, only random numbers, which are encrypted with a preloaded initial key, are exchanged among nodes. As the adversary is not preloaded with the initial key, it cannot compute the master keys. Because $K_{IN}$ and $RN_{i}$ are erased before the expiration of $T_{min}$, the capture of one node does not reveal the master keys of other nodes. The adversary can only access the two-hop broadcast keys stored on the capture node.Attacks against the authentication mechanism: one-way hash chain is largely used in authentication. An adversary can only extract the hash chain of the captured node, and hence it cannot generate observations using the identity of other nodes.Attacks using Fabrication: as a node does not have the two-hop broadcast keys of its neighbor, it cannot alter the observations it has to relay to the monitoring node. However, an adversary can make a captured node generate false observations. To deal with this problem, we use the solutions that are presented in Section 5.2.4 and Section 5.2.5.Beacon dropping attack: a malicious node might drop the beacon packets originated from honest nodes. As long as there is one path with only honest and well-behaved nodes, the fair rating will eventually reach the monitoring node.

### 6.2. Full-Resilience Probability Analysis

In this section, we provide the theoretical analysis of the full-resilience probability of the reputation system against selective routing attacks and unfair ratings.

We assume that *N* nodes are uniformly distributed in a region of area *A*. The probability density function (pdf) of the distance *S* between two nodes with a uniformly distributed node is given by [44]: $$f_{S}\left(s\right)=\frac{\hat{s}}{9\pi }\left[18\pi -36\arcsin\left(\frac{\hat{s}}{2}\right)-9\hat{s}\sqrt{4-{\hat{s}}^{2}}\right]$$
where $\hat{S}=\frac{S}{a}$, and $a=\sqrt{\frac{A}{\pi }}$.

We assume that all of the nodes have the same transmission radio range *r*. Two nodes establish a link if they are located within a distance of *r* from each other. The probability that the distance between two nodes is less than or equal to *r* is given by [45]: $$P_{1}=P\left(S\leq r\right)=\int_{0}^{r}f_{S}\left(s\right)ds$$

The necessary condition for a monitoring node *m* to correctly rate node *u* is $WN_{u}\neq Ø$, i.e., at least one node must reside in the region $R_{1}$ of area $A_{1}\left(d\right)$, which results from the intersection of two circular communication regions with radius *r* and centered at *u* and its child node ω. The distance *d* between the two nodes *u* and ω, must be in the range between 0 and *r*. The area $A_{1}\left(d\right)$ is calculated, as follows: $$A_{1}\left(d\right)=2r^{2}cos^{-1}\left(\frac{d}{2r}\right)-d\sqrt{r^{2}-\frac{d^{2}}{4}}$$

The expected area $E[A_{1}]$ is calculated as follows: $$E\left[A_{1}\right]=\int_{0}^{r}f_{S}\left(s\right)A_{1}\left(s\right)ds$$

The expected number of witness nodes located in $R_{1}$ is $N_{A1}=\frac{E[A_{1}]N}{A}$


A monitoring node *m* is connected to the witness node *w* either directly or through a relay node. The relay node must reside in the region $R_{2}$ of area $A_{2}\left(d\right)$, which is resulted from the intersection of two circular communication regions with radius *r* and centered at *m* and *w*. The area $A_{2}\left(d\right)$ is calculated, as follows: $$A_{2}\left(d\right)=2r^{2}cos^{-1}\left(\frac{d}{2r}\right)-d\sqrt{r^{2}-\frac{d^{2}}{4}}$$

The expected area $E[A_{2}]$ is calculated, as follows: $$E\left[A_{2}\right]=\int_{r}^{2r}f_{S}\left(s\right)A_{2}\left(s\right)ds$$

The expected number of relay nodes located in $R_{2}$ is $N_{A2}=\frac{E[A_{2}]N}{A}$


The probability that at least one relay node resides in the region $R_{2}$ is given by: $$I\left(d\right)=1-{\left(1-\frac{A_{2}\left(d\right)}{A}\right)}^{N-2}$$

The connection probability between the two nodes, which communicate in two hops and are separated by a distance *d* ranging between *r* and 2r, is calculated as follows [46]: $$P_{2}=\int_{r}^{2r}f_{S}\left(s\right)I\left(s\right)ds$$

We consider that a monitoring node can directly communicate with $N_{A1}^{1}$ witness nodes and with $N_{A1}^{2}=\left(N_{A1}-N_{A1}^{1}\right)$ witness nodes through relay nodes.

In Figure 5, the witness node *w* can reach the monitoring node *m* via the relay nodes *u*, $w_{1}$, and $w_{2}$. Nodes $w_{1}$ and $w_{2}$ also witness the traffic transmitted on $\left(u_{1},u\right)$ and $\left(u_{2},u\right)$, respectively.

The average probability that the monitoring node *m* is connected to all $N_{A1}$ witness nodes is given by: $$P_{fullcon}=P_{1}^{N_{A1}^{1}}P_{2}^{N_{A1}^{2}}$$

Let $P_{comp}$ be the probability that the node is compromised. The average probability that there is at least one path, which is free from compromised nodes, between one witness node and the monitoring node is: $$P_{sec}=\left\{\begin{array}{cc} P_{1}\times \left(1-P_{comp}\right) & one-hop\\ P_{2}\left(1-P_{comp}\right)\left(1-P_{comp}^{N_{A2}}\right) & two-hop \end{array}\right.$$

The average probability that there is at least one path, which is free from compromised nodes, between the majority of witness nodes and the monitoring node is: $$\left(\sum_{k=\lfloor \frac{N_{A1}}{2}\rfloor +1}^{N_{A1}}C_{k}^{N_{A1}}P_{sec}^{k}{\left(1-P_{sec}\right)}^{N_{A1}-k}\right)\times P_{fullcon}$$

## 7. Comparative Analysis: Resiliency against Upstream-Node Effect

In this section, we compare between the relaxed flow conservation constraint, the packet sending rate, and the packet dropping rate with respect to resiliency against the upstream-node effect, as shown in Figure 6. In this comparison analysis, we consider two nodes: *M*, and *P*, and *M* is sending packets to *P*. We consider the following notations:*x* is the number of packets dropped by the upstream node *M*.$PF_{P}$ denotes the number of packets that are forwarded by node *P*.$PR_{P}$ denotes the number of packets that are received by node *P* from node *M*.

**Proposition** **3.**
*The packet sending rate feature is not resilient against upstream-node effect.*


**Proof.** Node *P* is considered by a monitoring node *O* to be legitimate if $PR_{P}>Th_{SR}$. Otherwise, it is considered as malicious.A malicious node *M* drops *x* packets, and node *P* only receives and forwards $PR_{P}-x$ packets accordingly. Thus, the packet sending rate observed at node *P* is $(PR_{P}-x$). We will prove Proposition 3 by contradiction: let us assume that node *P* is always considered by a monitoring node as legitimate. Formally,
(1)$$\forall x:PR_{P}-x>Th_{SR}$$From (Equation 1), it follows that: $\forall x:f_{SR}\left(x\right)=PR_{P}-x-Th_{SR}>0$. As shown in Figure 6a, $\exists x:f_{SR}\left(x\right)\leq 0$, which contradicts Assumption Equation 1. Hence, for some values of *x*, node *P* is considered by a monitoring node as malicious. Therefore, the packet sending rate feature cannot prevent the upstream-node effect. □

**Proposition** **4.**
*The packet dropping rate feature is not resilient against upstream-node effect.*


**Proof.** Node *P* is considered by a monitoring node *O* as legitimate if $\left(1-\frac{PF_{P}}{PR_{P}}\right)\leq Th_{DR}$. Otherwise, it is considered to be malicious.A malicious node *M* drops *x* packets, and node *P* only receives and forwards $PR_{P}-x$ packets accordingly. Thus, the packet dropping rate observed at node *P* is $\left(1-\left(\frac{PF_{P}-x}{PR_{P}-x}\right)\right)$. We will prove Proposition 4 by contradiction: let us assume that node *P* is always considered by a monitoring node as legitimate. Formally,
(2)$$\forall x:\left(1-\left(\frac{PF_{P}-x}{PR_{P}-x}\right)\right)\leq Th_{DR}$$From (Equation 2), it follows that: $\forall x:f_{DR}\left(x\right)=\left(1-\left(\frac{PF_{P}-x}{PR_{P}-x}\right)-Th_{DR}\right)\leq 0$. As shown in Figure 6b, $\exists x:f_{DR}\left(x\right)>0$, which contradicts Equation (Equation 2). Hence, for some values of *x*, node *P* is considered by a monitoring node as malicious. Therefore, the packet dropping rate feature cannot prevent the upstream-node effect. □

**Proposition** **5.**
*The relaxed flow conservation feature is resilient against upstream-node effect.*


**Proof.** Node *P* is considered by a monitoring node *O* as legitimate if $\left(PR_{P}-PF_{P}\right)\leq Th_{RF}$. Otherwise, it is considered as malicious.A malicious node *M* drops *x* packets, and node *P* only receives and forwards $PR_{P}-x$ packets accordingly. Thus, the relaxed flow conservation feature observed at node *P* is $\left(PR_{P}-PF_{P}-x\right)$. We check whether the following statement is correct:
(3)$$\forall x:\left(PR_{P}-PF_{P}-x\right)\leq Th_{RF}$$From (Equation 3), it follows that: $\forall x:f_{RF}\left(x\right)=\left(PR_{P}-PF_{P}-x-Th_{RF}\right)\leq 0$, as shown in Figure 6c, and which proves Proposition 5. Hence, *P* is always considered by a monitoring node as legitimate. □

## 8. Performance Analysis

### 8.1. Complexity Analysis

In this section, we analyze the performance of the proposed intrusion detection system against selective routing attacks. The performance is studied under the following metrics: communication complexity and message complexity, which are defined as follows:Communication complexity (CC): The number of one-hop transmission each node needs to perform initially and periodically.Message complexity (MC): The size in terms of bits of control information transmitted by each node initially and periodically.

The complexity results are summarized in Table 1. Initially, each node *i* broadcasts $Witnessed_{i}$ with TTL=2, i.e., the messages are broadcast by *i* and its neighbors. This operation leads to a communication complexity of $O(d_{max}+1)$, where $d_{max}$ is the maximum node degree of the network. The same complexity is incurred by sending $RN_{i}$. By adding the broadcast operation of $Witnessing_{i}$, the total communication cost becomes proportional to $O(2d_{max}+3)$ per node. Let RN and TC be the size of $RN_{i}$ and TOHC value, respectively. As the size of $Witnessed_{i}$ and $Witnessing_{i}$ is $O(d_{max}\log_{2}N)$, the message complexity in the initialization phase is $O(\left(d_{max}^{2}+2d_{max}\right)\log_{2}N+\left(d_{max}+1\right)RN)$. The proposed IDS does not cause any additional packets during the periodic step, but it piggybacks some control information to the beacon packet. During the time period of $\left(|Relay_{i}|+2\right)$ slots, each node *i* relays the ratings of its neighbors in $Relay_{i}$ and its two own ratings. The periodic message complexity is $O(d_{max}+\log_{2}N)$, as explained in Section 5.2.2. In addition, the TOHC value is appended to each observation.

### 8.2. Energy Dissipation and Node Lifetime Analysis

We use the radio energy model, as that in [47]. The energy dissipation of transmitting (resp., receiving) one bit of information is given by: $E_{Tx}=E_{elec}+E_{amp}\times Dis^{\alpha }$ (resp., $E_{Rx}=E_{elec}$), where: Dis is the distance separating two nodes, and α is the attenuation factor of the environment and can be between 2 for free space and 4 for urban environment. The parameters that are used in [47] are as follows: $E_{elec}=50nJ/bit$, $E_{amp}=100pJ/bit/m^{2}$. Let D_Pkt and B_Pkt be the size in bit of data packet and beacon packet sent at each time period TB, respectively. The tree-based routing protocol with and without the proposed IDS is symmetric in the sense that every node executes the same code, i.e., every time that a node sends a message, it has to receive the same type of message from each neighbor node. Subsequently, the energy dissipated by a node under a tree-based routing protocol is: $$E_{W}=\left(D\_Pkt+B\_Pkt\right)\left(E_{Tx}+d_{max}E_{Rx}\right)$$

In the case of the same routing protocol with the proposed IDS, the energy dissipated by a node is composed of two terms: The first term is the energy dissipated during the initialization phase, and it is computed as: $$E_{IDS}^{a}=\left(\left(d_{max}^{2}+2d_{max}\right)\log_{2}N+\left(d_{max}+1\right)RN\right)\left(E_{Tx}+d_{max}E_{Rx}\right)$$

The second term, which is consumed periodically at each TB, is computed as: $$E_{IDS}^{b}=\left(D\_Pkt+B\_Pkt+d_{max}+\log_{2}N+TC\right)\left(E_{Tx}+d_{max}E_{Rx}\right)$$

Let the energy initial of each node be $E_{init}$. The node lifetime under the tree-based protocol is
$$T_{W}=\left(\frac{E_{init}}{E_{w}}\right)\times TB$$

The node lifetime under the IDS is: $$T_{IDS}=\left(\frac{E_{init}-E_{IDS}^{a}}{E_{IDS}^{b}}\right)\times TB$$

Let N=1024, $d_{max}=10$, RN=32, TC=32, and α=2. For the rest of parameters, we use the following, as in [48]: D_Pkt=4000 bytes, B_Pkt=20 bytes, Dis=50 m, and $E_{init}=3$J. Therefore, we obtain: $E_{W}=25.728mJ$, $E_{IDS}^{a}=1.241mJ$, $E_{IDS}^{b}=25.769mJ$, $T_{W}=116.60\times TB$, and $T_{IDS}=116.37\times TB$.

## 9. Simulation Results

In this section, we study the performance of the proposed intrusion detection system against selective routing attacks using GloMoSim simulator [49]. In our simulation scenarios, each malicious node launches a selective routing attack with probability *P* for each packet that it receives. To simulate a lossy wireless channel, we assume the Two-Ray path loss propagation model. In this model, the received power $P_{r}$ is modelled as: $P_{r}=P_{t}G_{t}G_{r}\frac{h_{t}^{2}h_{r}^{2}}{D^{4}}$ where: $P_{t}$ is the transmission power. $G_{t}$ and $G_{r}$ are antenna gains of transmitter and receiver respectively. $h_{t}$ and $h_{r}$ are the heights of both antennas. *D* is the distance between the transmitter and the receiver. The packet reception model is based on the signal to noise ratio (SNR) threshold. When the SNR is larger than a defined threshold, the signal is received without errors. Otherwise, the packet is dropped. The simulation parameters and their values are shown in Table 2. We generate the topology of the network using the NetLogo library [50], as shown in Figure 7.

Each node generates ten packets/second towards the sink, and each monitoring period lasts for 10 s. After a training phase of *T* time periods, testing phase lasts for 1800 s. The role of the IDS, which is implemented at each node *i*, is not just to detect if *i*’s neighbor (e.g., node *j*) is malicious or not, but also to know whether node *j* is malicious during a given time period. The dataset that is produced from the simulation can be found in [51]. We evaluate the performance of the IDS using the following six metrics:$$Accuracy=\frac{TP+TN}{TP+TN+FP+FN}$$
$$Precision=\frac{TP}{TP+FP}$$
$$Recall=\frac{TP}{TP+FN}$$
$$F-score=\frac{2(Precision\times Recall)}{Precision+Recall}$$
$$False\;positive\;rate=\frac{TP}{TP+FN}$$
$$False\;negative\;rate=\frac{FN}{TP+FN}$$
where TP, TN, FP, and FN denote the true positives, true negatives, false positives, and false negatives, respectively.

Figure 8a shows the values of $N_{Low}$ that are observed by a given node in the network while monitoring one of its neighbors. At lower training periods, $N_{Low}=99$ and it stabilizes at 98 starting from T=30. Figure 8b–e show the recall, precision, accuracy, and F-score of the proposed IDS, respectively, as a function of dropping probability. The first observation we can draw from the figure is that the recall is 100% when the dropping probability is higher than 0.05, and it is under 100% when the dropping probability becomes 0.02 and 0.01. This can be explained, as follows: under very low dropping probabilities, the malicious nodes perform at low intensities and their activities become unnoticeable. This happens when it is difficult to distinguish between packet loss due to normal activities and packet drop attack, and we can notice this when the dropping probability becomes very close or less than the normal packet loss, which is, at most, 2% during each time period. The same observation can be made from Figure 8c–e. However, they record lower results compared to recall results due to the effect of false positives. Figure 8f–h show the recall, false negative rate, and false positive rate of the IDS, respectively, as a function of the training period. The results are presented under the following levels of dropping probability P=1,0.5,0.1,0.05,0.01. Under high dropping probabilities, the recall (resp., false negative rate) is 100% (resp., 0%) for all of the training periods. Under low dropping probabilities, the detection rate decreases as the malicious behavior becomes very close to the normal one. We can notice that the false positive rate becomes 0 when the training period *T* reaches 30 in certain cases and 40 in others. At these values of the training period, $N_{Low}^{*}$ is observed and the IDS can accurately distinguish between normal traffic and selective routing attack. For instance, when the dropping probability is 0.5, the false positive rate becomes 0 at T=30. Before $N_{Low}^{*}$ is observed (i.e., T<30 or T<40), the false positive rate curves are random, as they depend on the number of times the monitored profile is observed inside the interval $\left[N_{Low}^{*},N_{Low}\right]$, which itself depends on the probability of packet loss.

## 10. Conclusions

In this paper, we have proposed an intrusion detection system against selective routing attack in wireless sensor networks. To counter a novel threat, named upstream-node effect, the proposed IDS employs a one-dimensional one-class classifier, named relaxed flow conservation constraint. We have derived four path-free features from the relaxed flow conservation constraint, in order to monitor all its neighbors. Three features can be obtained directly using one-hop information and the fourth one is obtained using the proposed two-hop energy-efficient and secure reporting scheme. We have analyzed the security of the proposed IDS and its resilience probability against the selective routing attacks and unfair ratings. We have also provided a performance complexity of the IDS and a comparison between the relaxed flow conservation and other features. The node lifetime analysis demonstrates that the energy consumption incurred by the IDS is insignificant. The simulation results show that the proposed intrusion detection system achieves good results in terms of detection effectiveness. It can achieve a recall of 100% when the dropping probability is higher than the normal packet loss rate, and it can also achieve a false positive rate of 0% when $N_{Low}^{*}$ is observed during the training phase. As a future work, we plan to relax some assumptions and consider different transmission rates. Additionally, it would be interesting to integrate the blockchain technology with the proposed scheme.

## Figures and Tables

**Figure 1 sensors-20-06106-f001:**
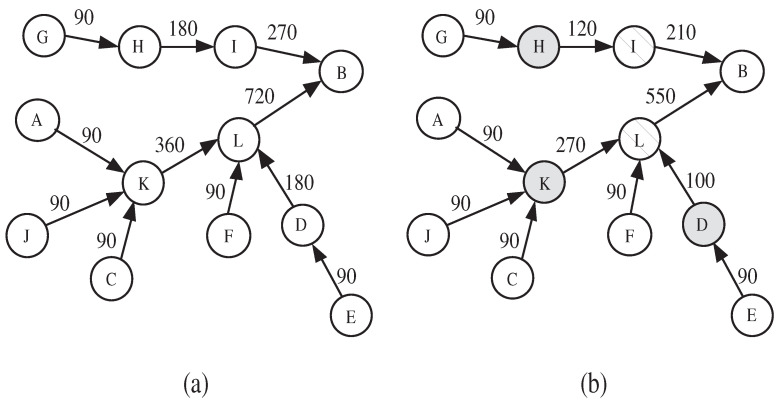
Impact of upstream-node effect on false positive rate: (**a**) normal sending rate, (**b**) sending rate under packet dropping attack.

**Figure 2 sensors-20-06106-f002:**
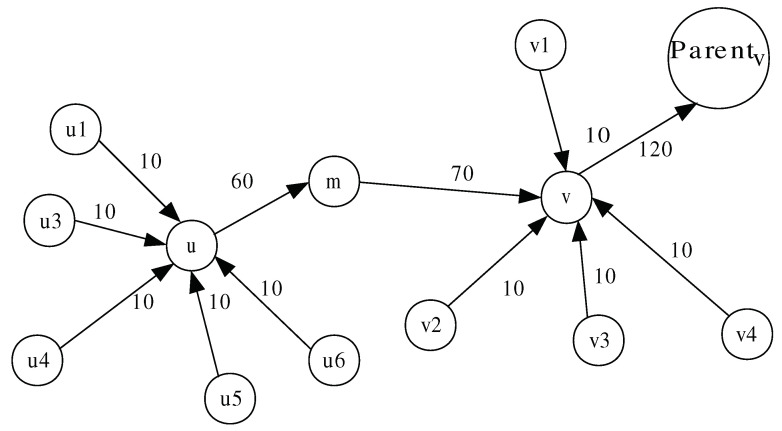
Flow conservation constraint.

**Figure 3 sensors-20-06106-f003:**
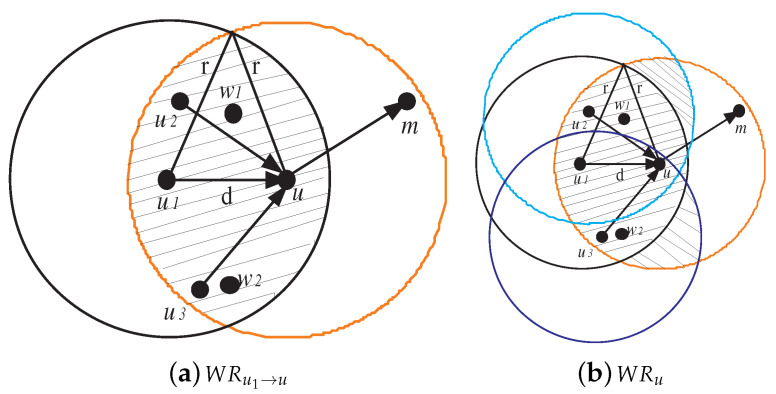
Witness region for $\left(u_{1},u\right)$ and total witness region for *u*.

**Figure 4 sensors-20-06106-f004:**
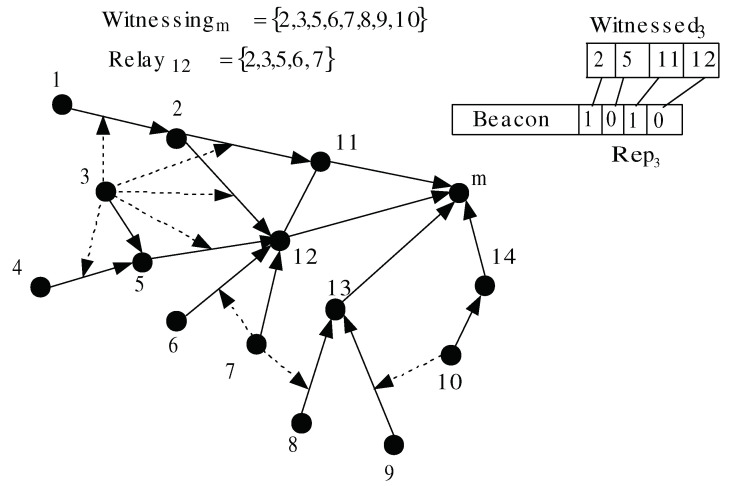
Initialization and rating phases.

**Figure 5 sensors-20-06106-f005:**
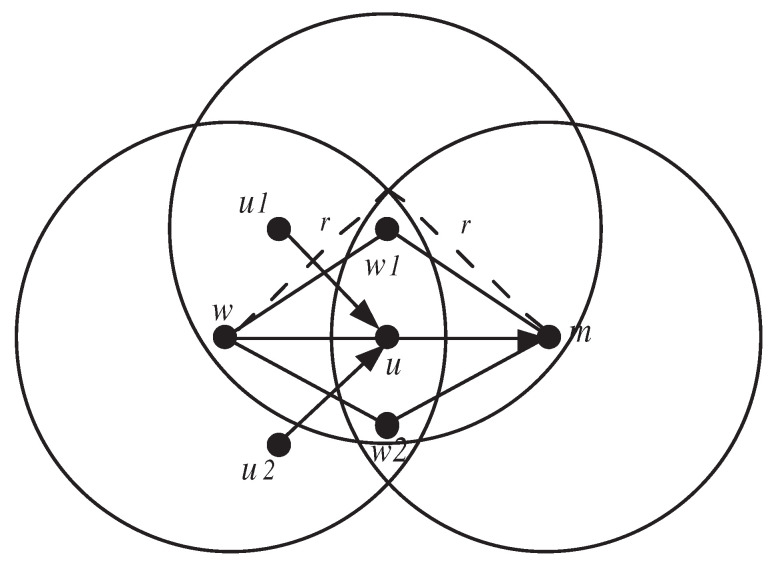
Communication between witness and monitoring nodes.

**Figure 6 sensors-20-06106-f006:**
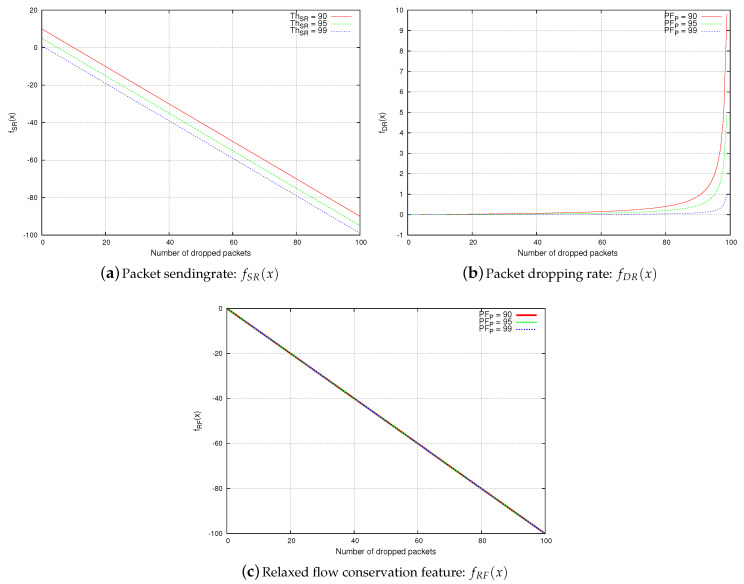
Resiliency against upstream-node effect.

**Figure 7 sensors-20-06106-f007:**
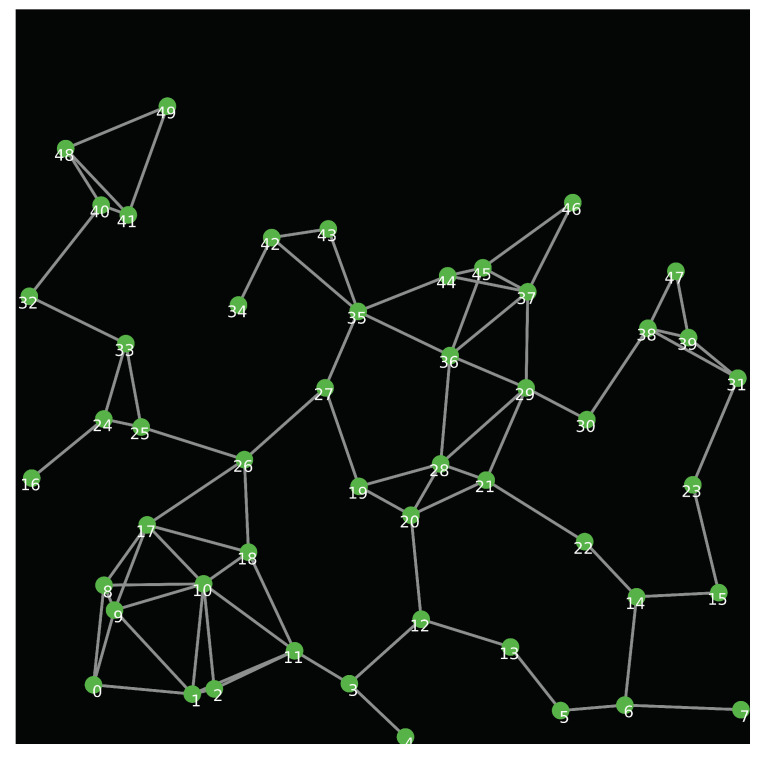
Network topology.

**Figure 8 sensors-20-06106-f008:**
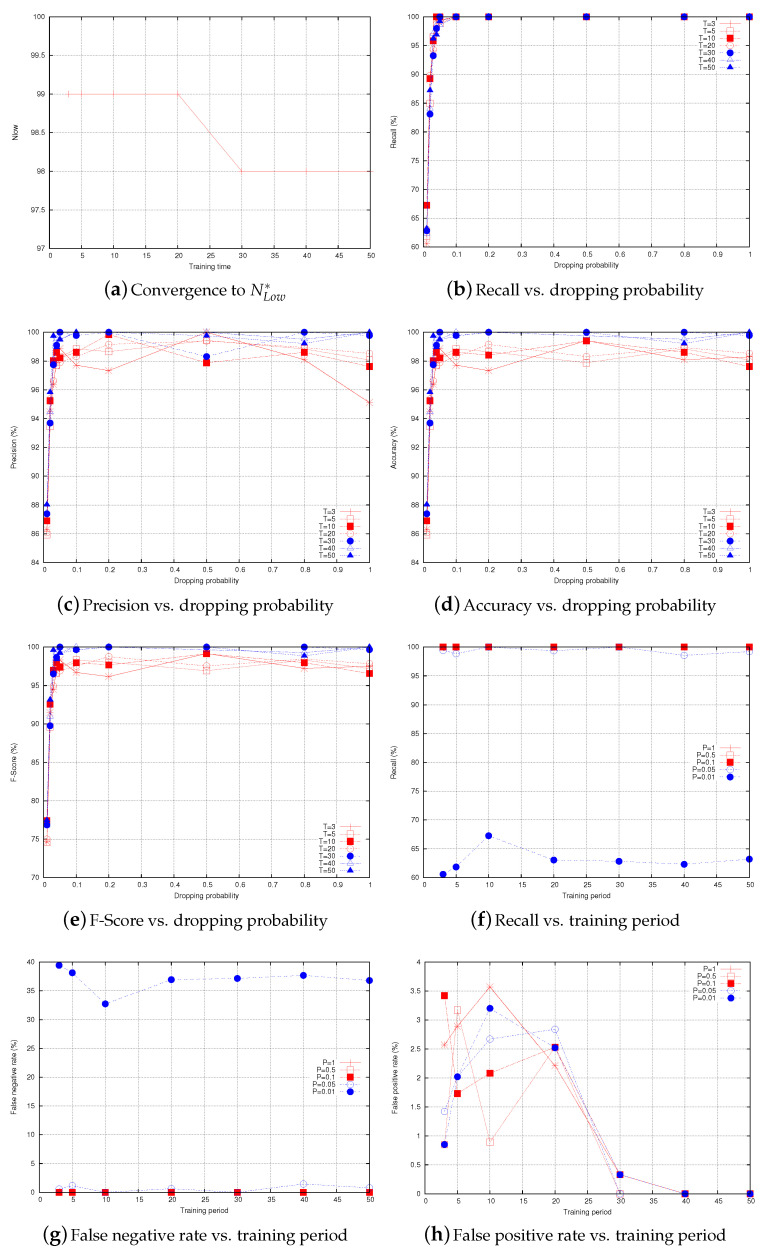
Performance of the proposed IDS.

**Table 1 sensors-20-06106-t001:** Complexity analysis of the proposed IDS.

CC (Initial)	$O(2d_{max}+3)$
CC (Periodic)	0
MC (Initial)	$O(\left(d_{max}^{2}+2d_{max}\right)\log_{2}N+\left(d_{max}+1\right)RN)$
MC (Periodic)	$O(d_{max}+\log_{2}N+TC)$

**Table 2 sensors-20-06106-t002:** Simulation Parameters.

**Parameters**	**Values**
Number of nodes	50
Dropping probability	1, 0.8, 0.5, 0.2, 0.1, 0.05, 0.04, 0.03, 0.02, 0.01
Monitoring period (Δ)	10 s
Training period (*T*)	3, 5, 10, 20, 30, 40, 50 ×Δ
Transmission power ($P_{t}$)	5 dBm
$G_{t}$, $G_{r}$	0 db
$h_{t}$, $h_{r}$	1.5 m
SNR Threshold	10 db

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
