# Peer review of "Two-Hop Monitoring Mechanism Based on Relaxed Flow Conservation Constraints against Selective Routing Attacks in Wireless Sensor Networks"

_sensors, 2020, doi:10.3390/s20216106_

Round 1

Reviewer 1 Report

In this paper, the authors have proposed a one-dimensional one-class classifier, named relaxed flow conservation constraint, as an intrusion detection scheme to counter a novel threat, named upstream-node effect.
It has been shown that relaxed flow conservation constraint prevents the upstream node effect contrary to the state-of-the-art methods. The proposed IDS has been validated through security analysis, performance complexity, and simulation experiments.

Overall, the paper is well-written, the proposed solution is well-described, and the research is thorough.

Some points that need to be considered to improve the paper are as follows:

- Authors should increase the size of the figures 1-5.

- Authors should replace the caption of figure 7 by: "Performance of the proposed IDS", and the caption of subfigure (a) by : "Convergence to N*low".

- In Section 4.2, the authors should remove "(resp., does not depend)" from the definition of "path-dependent feature" as it confuses the reader.

- The paper contains some typos (see below). The authors should proofread the paper to correct typos and grammatical mistakes such as:

Line 78 : In Section 7 ----> In section 7
Line 81 : Finally, Section 10 ---> Finally, section 10
Line 156: incur poor detection accuracy and false positive ---> poor detection accuracy and A high false positive rate.
Line 167: single-of-failure problem ---> single-POINT-of-failure
Line 184: After time period T ---> After a time period T
Line 203: with density lower ---> with A density lower
Line 554: the following metric--> the following metrics
Line 584: without error---> without errorS
Line 591: one of its neighbor ---> one of its neighbors
Line 591: it stabilizes to 98 ---> it stabilizes at 98

Author Response

We greatly appreciate the Editor-in-chief, the guest editors, and the reviewers for their insightful and valuable comments that greatly help us improve the quality of the manuscript. According to the reviewers’ comments, we made the following revisions to the manuscript. The changes are highlighted in yellow color.

  • We extended Section 5.2.4 to discuss and compare between the exogenous and the endogenous approaches with respect to message overhead cost, and justified the choice of the endogenous approach.
  • We extended Section 9 to evaluating the proposed IDS using additional metrics, namely:  Precision, Accuracy, F-Score, and false negative rate, which are presented in Figure 8-c, Figure 8-d, and Figure 8-e, and Figure 8-g respectively. 
  • We extended Section 9 to add the network topology (Figure 7) and add Table of simulation parameters.
  • We added labels to the axes of the graphs of Figure 6, and we put the correct curves in Figure 6-b.
  • Editing issues and typos were fixed.

Comment R1.1: Authors should increase the size of the figures 1-5.

Response to comment R1.1: The size of the figures was increased

Comment R1.2:  Authors should replace the caption of figure 7 by: "Performance of the proposed IDS", and the caption of subfigure (a) by : "Convergence to N*low".

Response to comment R1.2: The caption of the main figure and the sub-figure were updated according to the reviewer’s suggestion.

Comment R1.3:  In Section 4.2, the authors should remove "(resp., does not depend)" from the definition of "path-dependent feature" as it confuses the reader.

Response to comment R1.3: The sentence "(resp., does not depend)"  was  removed from the definition

Comment R1.4:  The paper contains some typos (see below). The authors should proofread the paper to correct typos and grammatical mistakes such as:

Line 78 : In Section 7 ----> In section 7
Line 81 : Finally, Section 10 ---> Finally, section 10
Line 156: incur poor detection accuracy and false positive ---> poor detection accuracy and A high false positive rate.
Line 167: single-of-failure problem ---> single-POINT-of-failure 
Line 184: After time period T ---> After a time period T
Line 203: with density lower ---> with A density lower
Line 554: the following metric--> the following metrics
Line 584: without error---> without errorS
Line 591: one of its neighbor ---> one of its neighbors
Line 591: it stabilizes to 98 ---> it stabilizes at 98

Response to comment R1.4: We proofread the paper and the highlighted typos were corrected.

Reviewer 2 Report

The authors present an interesting work on Two-hop Monitoring Mechanism based on Relaxed Flow Conservation Constraints. The presentation is well balanced between new material and well-known theory to support it.

I suggest enlarging text in charts-figures.

Author Response

We greatly appreciate the Editor-in-chief, the guest editors, and the reviewers for their insightful and valuable comments that greatly help us improve the quality of the manuscript. According to the reviewers’ comments, we made the following revisions to the manuscript. The changes are highlighted in yellow color.

  • We extended Section 5.2.4 to discuss and compare between the exogenous and the endogenous approaches with respect to message overhead cost, and justified the choice of the endogenous approach.
  • We extended Section 9 to evaluating the proposed IDS using additional metrics, namely:  Precision, Accuracy, F-Score, and false negative rate, which are presented in Figure 8-c, Figure 8-d, and Figure 8-e, and Figure 8-g respectively. 
  • We extended Section 9 to add the network topology (Figure 7) and add Table of simulation parameters.
  • We added labels to the axes of the graphs of Figure 6, and we put the correct curves in Figure 6-b.
  • Editing issues and typos were fixed.

Comment R2.1: I suggest enlarging text in charts-figures.

Response to comment R2.1: The size as well as the charts text have been both enlarged and they are more visible now.

Reviewer 3 Report

Some improvements are suggested in the sequel:

* The proposed approach generates a kind of trust model. It is suggested in a future extension of this work the inclusion of blockchain technology that might advance the scheme.

* Also, in future work, it is suggested the relaxation of some assumptions, as the one stated on p. 14, l. 503, regarding the same transmission rate.

* Because only two performance metrics are included, the incorporation of additional metrics, as the F-measure and the recall metric is suggested.

* Another suggestion is to upload the produced dataset on a repository (labelled), in order to have it freely available for other scholars to perform similar experiments.

* In the simulation section, a large number of parameters are introduced, which make the evaluation of the presented outcomes difficult. It is suggested that the authors add a table with all parameters and their corresponding values used. It would be beneficial to add the number of nodes that were used during simulations and the topology of the nodes (e.g. grid).

* In the introduction, among the main contributions discussed on p. 2, there are some that do not really contribute. It is suggested to reconsider these paragraphs.

* Figures are fade with tiny font. It is very hard to read them although some of them are colored.

Some suggestions for typos and grammar corrections follow:

* p. 1, l. 24, The sentence “Although … [4]” makes no sense.

* p. 7, l. 265, “,…”, incomplete sentence.

* p. 10, l. 362, “b)” is omitted.

* p. 12, l. 440, “the” is double.

* p. 14, l. 490: “One-way hash chain…”.

* p. 15, l. 512, “Figure 6.” has no discussion here.

Author Response

We greatly appreciate the Editor-in-chief, the guest editors, and the reviewers for their insightful and valuable comments that greatly help us improve the quality of the manuscript. According to the reviewers’ comments, we made the following revisions to the manuscript. The changes are highlighted in yellow color.

  • We extended Section 5.2.4 to discuss and compare between the exogenous and the endogenous approaches with respect to message overhead cost, and justified the choice of the endogenous approach.
  • We extended Section 9 to evaluating the proposed IDS using additional metrics, namely:  Precision, Accuracy, F-Score, and false negative rate, which are presented in Figure 8-c, Figure 8-d, and Figure 8-e, and Figure 8-g respectively. 
  • We extended Section 9 to add the network topology (Figure 7) and add Table of simulation parameters.
  • We added labels to the axes of the graphs of Figure 6, and we put the correct curves in Figure 6-b.
  • Editing issues and typos were fixed.

Comment R3.1: The proposed approach generates a kind of trust model. It is suggested in a future extension of this work the inclusion of blockchain technology that might advance the scheme.

 Response to comment R3.1:  In the conclusion section, we added this sentence “As future work, it would also be interesting to integrate the blockchain technology with the proposed scheme ”

Comment R3.2: Also, in future work, it is suggested the relaxation of some assumptions, as the one stated on p. 14, l. 503, regarding the same transmission rate.

Response to comment R3.2: In the conclusion, we also added this sentence: “As future work, we plan to relax some assumptions and employ different transmission rates”.

Comment R3.3: Because only two performance metrics are included, the incorporation of additional metrics, as the F-measure and the recall metric is suggested.

Response to comment R3.3: The proposed IDS has been evaluated using additional metrics, namely:  Precision, Accuracy, F-Score, and false negative rate, which are presented in Figure 8-c, Figure 8-d, and Figure 8-e, and Figure 8-g respectively.  In the revised version, we named detection rate as recall, and it is measured in Figure 8-b and Figure 8-f .

Comment R3.4: Another suggestion is to upload the produced dataset on a repository (labelled), in order to have it freely available for other scholars to perform similar experiments.

Response to comment R3.4: We uploaded the produced dataset to https://fac.ksu.edu.sa/abderhab/announcement/333117, and we cited it in the paper.

Comment R3.5: In the simulation section, a large number of parameters are introduced, which make the evaluation of the presented outcomes difficult. It is suggested that the authors add a table with all parameters and their corresponding values used. It would be beneficial to add the number of nodes that were used during simulations and the topology of the nodes (e.g. grid).

 Response to comment R3.5: Thank you for your valuable comment.

We added a table (Table 2) that presents simulation parameters and their values.

We also added the number of nodes in Table 2.

We also added a figure that shows the topology of the network.

Comment R3.6: In the introduction, among the main contributions discussed on p. 2, there are some that do not really contribute. It is suggested to reconsider these paragraphs.

Response to comment R3.6: We restructured the contribution points.

Comment R3.7: Figures are fade with tiny font. It is very hard to read them although some of them are colored.

Response to comment R3.7:  The size of the figures has been enlarged and they are now more visible

Comment R3.8: Some suggestions for typos and grammar corrections follow:

* p. 1, l. 24, The sentence “Although … [4]” makes no sense.

* p. 7, l. 265, “,…”, incomplete sentence.

* p. 10, l. 362, “b)” is omitted.

* p. 12, l. 440, “the” is double.

  • p. 14, l. 490: “One-way hash chain…”.

* p. 15, l. 512, “Figure 6.” has no discussion here.

Response to comment R3.8: We went through an extensive proofread and the highlighted typos have been addressed

Reviewer 4 Report

This paper is very well written and referenced. The authors have proposed the approach of a one-dimensional one-class classifier as a system for intrusion detection of the novel threat of an upstream-node effect.

I have some comments and recommendations.

  • Double check the paper for minor typos and missing words (e.g. line 247 - remove ‘but’ and ‘show’ should be ‘shown’).
  • In the final version, the outer parentheses in the equations should be larger than the inner parentheses in order to make the equations more clearly legible.
  • Although Figure 6 is discussed on page 15, the axes of the graphs should also be labeled.
  • The proofs of the Propositions are very important, and I’m glad the authors included them.
    The endogenous/exogenous tradeoff (i.e. exogenous considers the witness nodes’ reputations, etc.) is very important, and I’d like to see a little further discussion and specifics here about the message overhead cost associated with the exogenous method.

With the above minor corrections, I recommend this paper for publication.

Author Response

We greatly appreciate the Editor-in-chief, the guest editors, and the reviewers for their insightful and valuable comments that greatly help us improve the quality of the manuscript. According to the reviewers’ comments, we made the following revisions to the manuscript. The changes are highlighted in yellow color.

  • We extended Section 5.2.4 to discuss and compare between the exogenous and the endogenous approaches with respect to message overhead cost, and justified the choice of the endogenous approach.
  • We extended Section 9 to evaluating the proposed IDS using additional metrics, namely:  Precision, Accuracy, F-Score, and false negative rate, which are presented in Figure 8-c, Figure 8-d, and Figure 8-e, and Figure 8-g respectively. 
  • We extended Section 9 to add the network topology (Figure 7) and add Table of simulation parameters.
  • We added labels to the axes of the graphs of Figure 6, and we put the correct curves in Figure 6-b.
  • Editing issues and typos were fixed.

Comment R4.1: Double check the paper for minor typos and missing words (e.g. line 247 - remove ‘but’ and ‘show’ should be ‘shown’).

Response to comment R4.1: We went through an extensive proofread and the highlighted typos have been addressed

Comment R4.2: In the final version, the outer parentheses in the equations should be larger than the inner parentheses in order to make the equations more clearly legible.

 Response to comment R4.2: In the revised version, the outer parentheses in the equations  are now larger than the inner parentheses.

Comment R4.3: Although Figure 6 is discussed on page 15, the axes of the graphs should also be labeled

Response to comment R4.3: The axes labels have been added in the figure 6.

Comment R4.4: The proofs of the Propositions are very important, and I’m glad the authors included them.  The endogenous/exogenous tradeoff (i.e. exogenous considers the witness nodes’ reputations, etc.) is very important, and I’d like to see a little further discussion and specifics here about the message overhead cost associated with the exogenous method.

Response to comment R4.4: We added a discussion that compares between the exogenous and the endogenous approaches with respect to message overhead cost, and justified the choice of the endogenous approach.